# Inflammation-Related Biomarkers Are Associated with Heart Failure Severity and Poor Clinical Outcomes in Patients with Non-Ischemic Dilated Cardiomyopathy

**DOI:** 10.3390/life11101006

**Published:** 2021-09-24

**Authors:** Ieva Kažukauskienė, Vaida Baltrūnienė, Ieva Rinkūnaitė, Edvardas Žurauskas, Dalius Vitkus, Vytė Valerija Maneikienė, Kęstutis Ručinskas, Virginija Grabauskienė

**Affiliations:** 1Department of Pathology, Forensic Medicine and Pharmacology, Faculty of Medicine, Vilnius University, M. K. Čiurlionio 21, LT-03101 Vilnius, Lithuania; vaida.baltruniene@mf.vu.lt (V.B.); Edvardas.Zurauskas@vpc.lt (E.Ž.); virginija.grabauskiene@santa.lt (V.G.); 2Department of Biological Models, Life Sciences Center, Institute of Biochemistry, Vilnius University, 7 Saulėtekio av., LT-10257 Vilnius, Lithuania; ieva.rinkunaite@bchi.vu.lt; 3Department of Physiology, Biochemistry, Microbiology and Laboratory Medicine, Faculty of Medicine, Vilnius University, M. K. Čiurlionio 21, LT-03101 Vilnius, Lithuania; dalius.vitkus@santa.lt; 4Clinic of Cardiac and Vascular Diseases, Institute of Clinical Medicine, Faculty of Medicine, Vilnius University, M. K. Čiurlionio 21, LT-03101 Vilnius, Lithuania; vyte.maneikiene@santa.lt (V.V.M.); Kestutis.Rucinskas@santa.lt (K.R.)

**Keywords:** biomarkers, endomyocardial biopsy, inflammation, non-ischemic dilated cardiomyopathy, prognosis

## Abstract

Inflammation-related biomarkers are associated with clinical outcomes in mixed-etiology chronic heart failure populations. Inflammation-related markers tend to be higher in ischemic than in non-ischemic dilated cardiomyopathy (NI-DCM) patients, which might impact their prognostic performance in NI-DCM patients. Therefore, we aimed to assess the association of inflammation-related biomarkers with heart failure severity parameters and adverse cardiac events in a pure NI-DCM patient cohort. Fifty-seven patients with NI-DCM underwent endomyocardial biopsy. Biopsies were evaluated by immunohistochemistry for CD3+, CD45ro+, CD68+, CD4+, CD54+, and HLA-DR+ cells. Blood samples were tested for high-sensitivity C-reactive protein (hs-CRP), interleukin-6, tumor necrosis factor-α (TNF-α), soluble urokinase-type plasminogen activator receptor and adiponectin. During a five-year follow-up, twenty-seven patients experienced at least one composite adverse cardiac event: left ventricle assist device implantation, heart transplantation or death. Interleukin-6, TNF-α and adiponectin correlated with heart failure severity parameters. Patients with higher levels of interleukin-6, TNF-α, adiponectin or hs-CRP, or a higher number of CD3+ or CD45ro+ cells, had lower survival rates. Interleukin-6, adiponectin, and CD45ro+ cells were independently associated with poor clinical outcomes. All patients who had interleukin-6, TNF-α and adiponectin concentrations above the threshold experienced an adverse cardiac event. Therefore, a combination of these cytokines can identify high-risk NI-DCM patients.

## 1. Introduction

Heart failure is a multi-etiologic clinical syndrome that affects up to 2% of the adult population in developed countries [1]. One of the leading causes of heart failure is non-ischemic dilated cardiomyopathy (NI-DCM), affecting approximately one-third of heart failure patients [2]. Due to its etiopathogenetic diversity, the progression of NI-DCM varies between individuals. Despite the use of guideline-directed treatment, a significant proportion of NI-DCM patients deteriorate (progressively), either until death or until they receive a heart transplant. NI-DCM is the most common indication for heart transplantation worldwide, accounting for more than 50% of all heart transplants in patients younger than 60 years [3]. However, there are no prognostic strategies for risk stratification in patients who have developed NI-DCM. Furthermore, Dziewiecka et al. showed that most risk assessment models, which have been created for heterogeneous heart failure populations, have suboptimal accuracy for NI-DCM patients [4]. Therefore, there is a need for biomarkers that can aid in risk assessment and help identify disease progression, as well as to facilitate the search for specific therapeutic targets.

Infiltration of inflammatory cells in the myocardium and concurrent systemic inflammation are important factors in the development and progression of chronic heart failure [5]. Various inflammation-related biomarkers can predict poor clinical outcomes in the chronic heart failure population [6]. However, this population consists of multi-etiologic diseases, primarily NI-DCM and ischemic heart disease. Ischemic heart disease patients stand out not just because of a poorer prognosis than patients with non-ischemic heart failure [7,8] but also due to a higher expression and concentration of inflammation-related biomarkers [9,10,11]. Thus, the prognostic role of inflammation-related biomarkers in chronic heart failure studies might be primarily determined by patients with ischemic etiology. Although various studies demonstrate a distinct role for inflammation-related biomarkers in the pathogenesis of NI-DCM, there is a lack of data on the predictive value of such markers in NI-DCM patients. Therefore, we aimed to assess the association of inflammation-related myocardial and serum biomarkers with parameters of heart failure severity, and evaluate their predictive potential in a pure NI-DCM patient cohort.

## 2. Materials and Methods

### 2.1. Study Population and Protocol

A prospective cohort study with retrospective analysis was done in the Vilnius University Hospital Santaros Klinikos. We enrolled 57 patients with suspected NI-DCM that were admitted to the hospital for diagnostic evaluation between January 2010 and December 2013. The median duration of heart failure symptoms before enrollment was 12 (5–60) months. The inclusion criteria were symptoms and signs of heart failure, with echocardiographic evidence of left ventricular dilation and reduced left ventricular ejection fraction (LVEF) (≤45%). The main goal of the study was to look for etiopathogenetic factors in NI-DCM, such as cardiotropic viruses and myocardial inflammation, by analyzing various biomarkers in serum and endomyocardial biopsies [12]; and to find the differences in pathogenesis between non-inflammatory and inflammatory NI-DCM [13], which might have an impact on future research or the patient treatment process.

Exclusion criteria were:
Significant coronary artery disease, defined as at least 50% proximal stenosis of a coronary artery, or a history of myocardial infarction;Other causes of heart failure, such as heart muscle or primary valvular disease, hypertensive heart disease, advanced chronic kidney disease, endocrine disease, alcohol or drug abuse;Acute myocarditis (onset in the previous three months), or acute coronary syndrome as suspected by clinical presentation or diagnostic evaluation.

Study patients underwent a detailed medical interview, physical examination and routine laboratory tests, including complete blood count, high sensitivity C-reactive protein (hs-CRP), creatinine (CKD-EPI creatinine equation was used to estimate glomerular filtration rate (GFR)), high-sensitivity troponin T (hs-troponin T) and B-type natriuretic peptide (BNP). All patients underwent standard transthoracic echocardiography to obtain conventional echocardiographic measurements within 24 h before interventional procedures. The methodology has been described in detail elsewhere [14], except severe LV diastolic dysfunction, which matched grade III according to recommendations [15]. Coronary angiography was performed to exclude coronary artery disease. Right heart catheterization was performed as described previously [14]. In brief, the procedure was carried out for cardiac pressures (pulmonary capillary wedge pressure (PCWP), mean pulmonary arterial pressure (mPAP)) and cardiac index assessment, followed by endomyocardial biopsy from the interventricular septum.

All patients were treated according to the guidelines of the European Society of Cardiology [16] and provided informed consent. Ethical approval was obtained from the local Lithuanian Bioethics Committee (license numbers 158200-382-PP1-23; 158200-09-382-l03; and 158200-17-891-413).

### 2.2. Endomyocardial Biopsy

Endomyocardial biopsy procedure, storage of the biopsy samples, histological and immunohistochemical analyses were performed as described previously [12]. In brief, we detected infiltrative inflammatory cells in the myocardium using the following antibodies: T-lymphocyte CD3 (Agilent DAKO, Hamburg, Germany), active-memory T-lymphocyte CD45ro (Agilent DAKO, Hamburg, Germany), macrophage CD68 (Agilent DAKO, Hamburg, Germany), T-helper cell CD4 (Agilent DAKO, Hamburg, Germany), intracellular adhesion molecule 1 (ICAM-1) CD54 (Leica Biosystems, Newcastle, United Kingdom) and MHC class II cell surface receptor HLA-DR (Agilent DAKO, Hamburg, Germany). Positive cells were registered by an experienced pathologist and expressed as the number of cells per mm^2^. Three endomyocardial biopsy procedures were discontinued because of arrhythmias or right ventricular perforation, and as a result, immunohistochemical analysis was not performed for two of these patients due to a lack of biopsy material.

### 2.3. Biochemical Assays of Serologic Inflammation-Related Markers

Plasma samples were stored at −80 °C until analysis. The proinflammatory serum cytokines TNF-*α* and IL-6 were measured by solid-phase, chemiluminescent immunometric assays using IMMULITE/Immulite 1000 systems (Immulite, Siemens) according to the manufacturer’s instructions: TNF-*α* (Catalog number LKNFZ (50 tests) and LKNF1 (100 tests)), IL-6 (Catalog number LK6PZ (50 tests) and LK6P1 (100 tests)) and expressed as pg/mL. Adiponectin was measured using the Millipore Adiponectin assay according to the manufacturer’s recommendations (Millipore, Burlington, MA, USA) and expressed as mg/mL. Levels of human soluble urokinase-type plasminogen activator receptor (suPAR) were estimated by ELISA assay according to the manufacturer’s recommendations (Abbexa). Absorbance was measured at 450 nm with a spectrophotometer (Varioskan^®^Flash, Thermo Fisher Scientific, Vantaa, Finland). Final concentrations of suPAR are expressed as ng/mg protein.

### 2.4. Follow-Up

Patients were followed up for five years after enrollment in the study. The clinical outcome measure was defined retrospectively and was a composite endpoint of left ventricle assist device implantation, heart transplantation, or cardiovascular death. The time of the first event was included in the analysis. Adverse cardiac events were confirmed by medical records, national death registry records, or telephone interviews with the patients’ families. The clinical relevance of choosing these outcomes as a composite endpoint was based on the idea that all these outcomes are clinically relevant and reflect the same clinical and probably pathophysiological state of advanced heart failure when there are no more viable alternatives of treatment.

### 2.5. Statistical Analysis

Data analysis was performed using the R studio package (4.0.3 version). A *p*-value of <0.05 was considered statistically significant. Continuous variables are expressed as the median (25th percentile, 75th percentile) and categorical data as counts and percentages. The Mann–Whitney U test compared continuous variables between two groups, Kruskal-Wallis test—between three groups. Categorical variables were compared between the groups by the chi-square test or Fisher’s exact test if expected values were <5. The association between inflammation-related and heart failure severity parameters was assessed using the Spearman correlation.

The receiver operating characteristic (ROC) curve was used to identify the optimal cut-off value for the outcome prediction of each inflammation-related biomarker and to estimate their accuracy for predicting composite outcome measures. Kaplan–Meier analysis was used to compare the cumulative survival rates between the two groups of NI-DCM patients stratified by cut-off values of each inflammation-related biomarker. The log-rank statistic was used to evaluate the statistical significance of differences between the curves. Univariate Cox proportional hazards regression analysis was run for all baseline variables to evaluate their association with poor composite outcomes. All variables with a *p*-value < 0.1 in the univariate analysis were included in multivariate Cox regression analysis, which was performed using the stepwise backward elimination.

## 3. Results

### 3.1. Baseline Patient Characteristics

The baseline characteristics of our cohort are shown in Table 1 and Table 2. The median (25th percentile, 75th percentile) age of the patients was 47 (44–53) years, and 45 (79%) were men. The majority (90%) were classified as New York Heart Association (NYHA) functional class III-IV. Patients tended to have elevated cardiac pressures and impaired cardiac index, as well as elevated levels of BNP, hs-troponin T and TNF-α.

Twenty-seven patients experienced adverse cardiac events during the five-year follow-up:

Ten (18%) patients died;

Nine (16%) underwent heart transplantation (the urgency status according to heart allocation policy [17]: six patients had status 1–3 and three patients had status 4);

Eight (14%) had an LVAD implantation (seven patients had 1–3 INTERMACS profile [18], one patient had profile 4).

The other patients remained on conventional medical heart-failure therapy. Based on these outcomes, we divided the cohort into two groups: event (*n* = 27) and event-free (*n* = 30) groups. Patients in the event group tended to have a longer duration of heart failure symptoms before enrollment than patients in the event-free group, but the difference was not statistically significant. Patients who experienced adverse outcomes had significantly lower systolic blood pressure, more impaired LV function, and enlarged right ventricle (Table 1). In addition, there were higher levels of BNP and inflammation-related cytokines (IL-6, TNF-α and adiponectin) in the event group compared to patients that remained event-free (Table 2).

### 3.2. Association between Inflammation-Related Biomarkers and Heart Failure Severity

We evaluated the correlations between inflammation-related biomarkers and parameters that reflect heart failure severity (Table 3). WBC, suPAR, infiltrative CD68+, CD4+, CD54+, and HLA-DR+ cells did not correlate with any of the heart failure severity parameters. A weak but significant positive correlation was found between infiltrative CD3+ and CD45ro+ cells and BNP levels, while hs-CRP significantly but weakly correlated with LVEF and BNP. Furthermore, levels of IL-6, TNF-α and adiponectin increased according to NYHA class (Figure 1) and correlated with all heart failure severity parameters (LVEF, mPAP, PCWP, and BNP), except TNF-α, which did not correlate with PCWP. Finally, IL-6 and adiponectin levels correlated most strongly with parameters of heart failure severity, particularly BNP.

Then we evaluated the correlations between inflammatory cells in the myocardium and inflammation-related biomarkers in serum (Table 4). Inflammatory cells did not correlate with biomarkers of systemic inflammation, except moderate correlation between CD68+ cells/mm^2^ and IL-6.

### 3.3. Inflammation-Related Biomarkers and Risk Prediction

We performed ROC analysis to identify the best cut-off value for each inflammation-related biomarker for predicting outcomes. Cut-off values are presented in Appendix A. IL-6, TNF-α and adiponectin predicted adverse cardiac events with the highest accuracy (AUC 0.77, 0.65 and 0.70, respectively).

Kaplan–Meier survival analyses with log-rank tests were subsequently performed to reveal alterations in survival probability among patient groups based on the obtained cut-off values (Figure 2). Higher levels of inflammation-related markers (hs-CRP > 4.6 µg/mL, IL-6 > 4.53 pg/mL, TNF-α > 7.81 pg/mL, adiponectin > 17.14 μg/mL) and higher numbers of infiltrative inflammatory cells (CD3+ > 13 cells/mm^2^ and CD45ro+ > 11.5 cells/mm^2^) were associated with lower survival rates (Figure 2). No significant differences in survival probability were obtained for the other inflammation-related biomarkers.

Univariate Cox regression analysis showed that systolic and diastolic blood pressure, echocardiographic parameters (LVEF, severe LV diastolic dysfunction, RV end-diastolic diameter, severe RV dysfunction), right heart catheterization measurements (mPAP, PCWP), inflammation-related serum biomarkers (hs-CRP > 4.62 µg/mL, IL-6 > 4.53 pg/mL, TNF-α > 7.81 pg/mL and adiponectin > 17.14 μg/mL) as well as inflammatory CD3+ > 13 cells/mm^2^ and CD45ro+ > 11.5 cells/mm^2^ in the myocardium predicted adverse cardiac events. Parameters with a *p*-value < 0.1 in univariate Cox regression analysis are shown in Table 5. Multivariate Cox analysis revealed that IL-6 > 4.53 pg/mL, adiponectin > 17.14 μg/mL, hs-troponin T, CD45ro+ > 11.5 cells/mm^2^ and diastolic blood pressure could independently predict adverse cardiac events.

### 3.4. Combined Assessment of Inflammation-Related Cytokines

As all three cytokines that were measured predicted adverse cardiac events with the highest accuracy (Appendix A), they were subjected to further analysis. Figure 3 shows a risk stratification based on the three cytokines. All patients (*n* = 12) with IL-6 < 4.53 pg/mL, adiponectin < 17.14 μg/mL and TNF < 7.81 pg/mL had a 100% five-year event-free survival, whereas all patients (*n* = 10) with concentrations of these three cytokines above the cut-off values experienced adverse cardiac events. Patients with one or two cytokines above the cut-off value had a similar five-year survival: 50% and 40% event-free survival rate, respectively.

## 4. Discussion

This study evaluates the association of inflammation-related biomarkers with heart failure severity, as well as their prognostic value, in a cohort of NI-DCM patients. The main findings are that:

hs-CRP is associated with certain heart failure severity parameters (LVEF, BNP) and adverse cardiac events;

suPAR is not associated with heart failure severity or poor clinical outcomes in NI-DCM patients;

Higher counts of CD3+ T lymphocytes and CD45ro+ memory T cells correspond to a poorer clinical outcome.

Higher levels of inflammation-related cytokines (IL-6, TNF-α, adiponectin) are associated with heart failure severity and predict poor clinical outcomes.

An increase in serum levels of all three cytokines (IL-6 > 4.53 pg/mL, TNF-α > 7.81 pg/mL and adiponectin > 17.14 mg/mL) can be helpful in identifying high-risk patients.

hs-CRP is an acute-phase protein produced mainly by hepatocytes under the influence of cytokines. Elevated hs-CRP levels are detected and associated with poor clinical outcomes in chronic heart failure patients [19,20]. However, the predictive value of hs-CRP in NI-DCM patients has not been resolved. Lamblin et al. [21] evaluated hs-CRP predictive value in 546 patients with heart failure with reduced ejection fraction (non-ischemic etiology ~60%). hs-CRP was a predictor of mortality in the whole cohort and ischemic heart failure subgroup during the median 2.7-year follow-up but not in non-ischemic patients. Li et al. [22] found that hs-CRP was an independent predictor of mortality during an average of 2.6 years of follow-up in 622 NI-DCM patients. Ishikawa et al. [23]. also demonstrated that hs-CRP was an independent predictor of poor clinical outcome in a cohort of 84 NI-DCM patients during 42 months of follow-up. In our study, which employed a longer follow-up period than previous studies, we found that hs-CRP was associated with long-term adverse cardiac events, although it did not remain a significant predictor following multivariate analysis.

suPAR is a proinflammatory marker originating from proteolytic cleavage and releasing of the membrane-bound urokinase-type plasminogen activator receptor from vascular endothelial and immune cells [24]. It is associated with infectious diseases, systemic inflammation, malignancies [25,26,27] and cardiovascular diseases, predominantly ischemic heart disease [28]. Several studies have analyzed suPAR in chronic heart failure patients, but the results have been inconsistent. Lichtenauer et al. [29] found no difference in suPAR concentrations between NI-DCM (*n* = 65) and ischemic heart failure (*n* = 59) patients. However, they observed an increase in suPAR according to NYHA stage, which reached a plateau at NYHA stage III. In another study, Koller et al. [30] found that suPAR was associated with adverse cardiac events in 319 chronic heart failure patients (55% NYHA class II, 43% class III). van den Berg et al. [31] found suPAR was associated with adverse cardiac events in 263 mixed-etiology chronic heart failure patients (26% had NYHA III-IV class) during a median of 2.2 years of follow-up. In contrast to earlier findings, we did not detect any association between suPAR and heart failure severity parameters or adverse cardiac events. A possible explanation for this might be that most patients in our cohort were classified as NYHA III-IV when suPAR levels would be expected to reach a plateau [29]; consequently, this marker possibly loses its prognostic value in our more advanced NI-DCM patient cohort.

The presence of infiltrative inflammatory cells in the myocardium can be assessed to diagnose inflammation [32]. However, only a few studies have estimated the prognostic value of distinct inflammatory cells, with inconsistent results [33,34]. Zuern et al. [33] found no association between CD3+ and CD68+ cells and clinical outcome in 227 patients with congestive heart failure (79% chronic myocarditis or non-inflammatory DCM) during three-years of follow-up. Nakayama et al. [34] evaluated CD3+ and CD68+ cells in the myocardium in 182 DCM patients. During an average of 6.9 years of follow-up, both inflammatory cell types were associated with poor clinical outcomes but did not remain significant in a multivariate approach. Our results demonstrate that CD3+ and CD45ro+ cells, but not CD68+, CD4+, CD54+ or HLA-DR+ cells, were associated with poor clinical outcomes during the five-year follow-up period. Higher counts of CD45ro+ cells remained an independent predictor of adverse cardiac events in the multivariate analysis. These divergent results might be influenced by differences in statistical analysis and follow-up duration, unequal proportions of different etiopathogenetic phenotypes in the NI-DCM cohorts [12], as well as biopsy collection occurring at different time points over the course of the disease [35].

Adiponectin is an adipocyte-derived cytokine, which acts in obesity-linked diseases [36], renal failure, and various cardiovascular diseases [37]. However, its actions are multifaceted and controversial [38]. Adiponectin has cardioprotective [39], anti-atherogenic [40] and anti-inflammatory effects [41,42] in the context of atherosclerotic cardiovascular diseases. Despite these beneficial effects, adiponectin levels increase with the clinical worsening of chronic heart failure [43,44], and are associated with higher mortality [45,46]. Our study results support previous findings demonstrating an association of adiponectin with heart failure severity parameters and adverse clinical outcomes in NI-DCM patients. Previously, Wojciechowska et al. [47] found that elevated adiponectin levels were associated with poor clinical outcomes in 129 NI-DCM patients during three-years of follow-up. Our results are consistent with these findings and show that adiponectin remains a significant predictor in NI-DCM patients during the five-year follow-up.

IL-6 and TNF-α are proinflammatory cytokines produced mostly by activated monocytes and macrophages. The proinflammatory role of these cytokines in systemic inflammation is well established in various pathologies, including cardiovascular diseases such as atherosclerotic cardiovascular disease and chronic heart failure [48,49]. In addition, the prognostic role of IL-6 and TNF-α in chronic heart failure patients has been demonstrated in previous studies [50,51]. However, approximately half of these cohorts were composed of ischemic heart failure patients, which might have had a significant impact on the predictive value of these cytokines, considering that serum concentrations of IL-6 and TNF-α are significantly higher in ischemic heart failure patients than in patients with NI-DCM [9,10]. To the best of our knowledge, the prognostic value of IL-6 and TNF-α has not previously been estimated in a NI-DCM patient population. We found an association between the cytokines, IL-6 and TNF-α, and adverse cardiac events during the five-year follow-up period. In addition, IL-6 > 4.53 pg/mL was an independent predictor of long-term poor clinical outcomes.

The present study also revealed that inflammation-related cytokines (IL-6, TNF-α and adiponectin) were associated with heart failure severity. These results are consistent with those of previous studies, which showed an association between circulating cytokines and heart failure progression (i.e., NYHA functional class and/or BNP levels) [9,44,52]. Additionally, our results supplement and consolidate previous findings, in which, besides having an association with NYHA and BNP, cytokines also correlated with other heart failure severity parameters, such as LVEF, PCWP (except TNF-α), and mPAP. Moreover, these cytokines showed the highest accuracy in predicting clinical outcomes out of all the inflammation-related biomarkers that we investigated. Our most important clinically relevant finding was that the combination of all three cytokines could help identify high-risk patients. Patients who had low concentrations of all three biomarkers did not experience adverse cardiac events. In contrast, all patients with higher concentrations than the cut-off values of all three cytokines experienced adverse cardiac events during follow-up, mainly within the first two years. Therefore, the use of inflammation-related cytokines might aid in identifying high-risk NI-DCM patients, which could lead to changes in follow-up intensity, the timing for device therapy, or heart transplantation.

Overall, this research evaluated both biomarkers of myocardial and systemic inflammation. A significant finding to emerge from this study is that inflammatory cells in the myocardium did not correlate with biomarkers of systemic inflammation, except CD68+ cells with IL-6. There is a lack of consistent data about inflammatory cells in an advanced disease phase [53] because they are mainly studied in patients with acute myocarditis, but not in a more advanced disease phase–NI-DCM. Therefore, we hypothesize that the absence of the association may partly be explained by the existence of different etiopathogenetic sub-groups in this cohort with different myocardial inflammation statuses [12]. The other possible explanation for this might be that pathological processes in the myocardium, including inflammation, cause malfunction and remodeling of the heart, which triggers a systemic inflammatory response that proceeds to a low-grade inflammation (para-inflammation) state leading to further disease progression [54]. This hypothesis might be supported by another significant finding from this study: systemic inflammation-related biomarkers (cytokines) predicted clinical outcomes more significantly and with higher accuracy than infiltrative inflammatory cells. This finding also reflects the importance of chronic low-grade systemic inflammation in the disease progression.

### Limitations

Our study’s main limitation is the single-center small sample size, which prevents us from making generalizations about independent predictors of poor clinical outcomes or exact cut-off values for each inflammation-related biomarker in patients with NI-DCM. Small-sample size also limited deeper subgroup analysis (e.g., urgent versus non-urgent heart transplantation or LVAD implantation) and the ability to perform more extent survival analysis—competing risks regression models. In addition, we did not evaluate the dynamics of the biomarkers during follow-up, which could help to understand the pathologic processes leading to heart failure progression more comprehensively. Although we demonstrated the ability of risk stratification by three inflammation-related cytokines, a larger study is needed to verify and build on the findings of the present study.

## 5. Conclusions

Increased myocardial and systemic inflammation is associated with poor long-term outcomes. IL-6, adiponectin and memory T cells appear to be independent predictors of poor clinical outcomes. IL-6, TNF-α and adiponectin are associated with disease severity, and a combination of these can be helpful in identifying high-risk NI-DCM patients. Future larger-sample research is necessary to confirm and validate these findings.

## Figures and Tables

**Figure 1 life-11-01006-f001:**
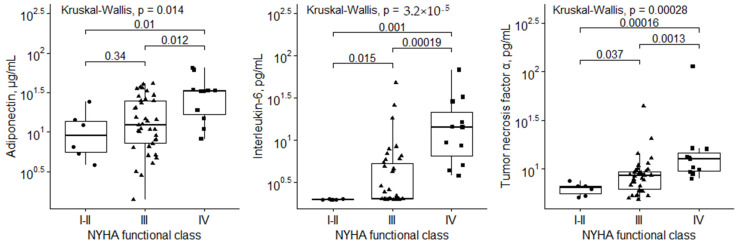
Serum cytokine levels based on NYHA functional class.

**Figure 2 life-11-01006-f002:**
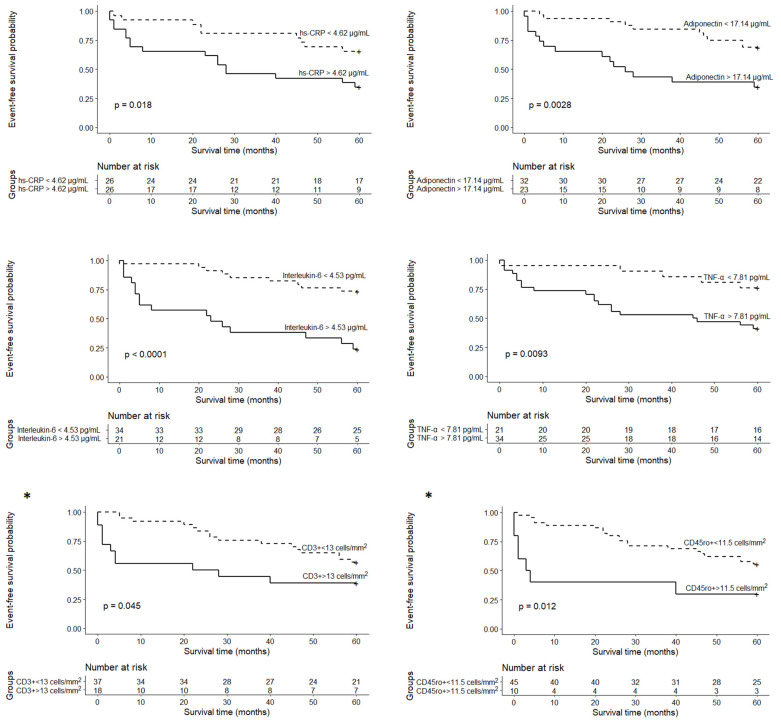
Survival curves stratified by inflammation-related biomarkers during the five-year follow-up. hs-CRP—high-sensitivity C reactive protein, TNF-α—tumor necrosis factor α. * Adapted with permission from https://doi.org/10.5603/cj.a2020.0088 (accessed on 19 September 2021) [12].

**Figure 3 life-11-01006-f003:**
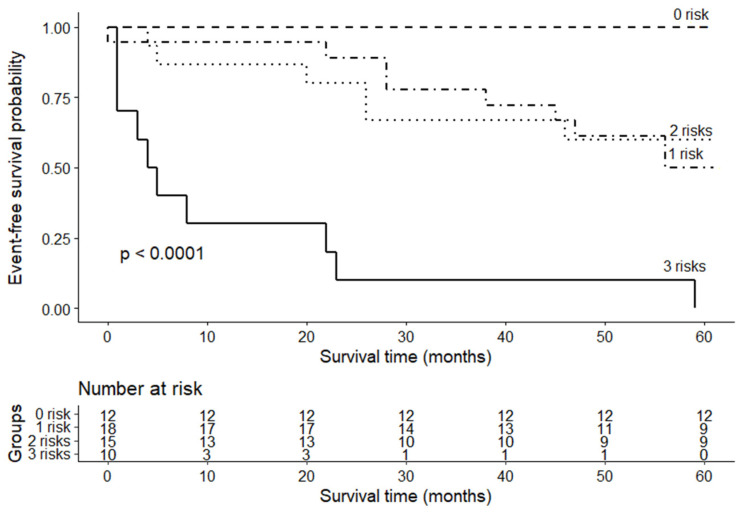
Event-free survival from adverse cardiac events according to a triple in Figure > 17.14 μg/mL, IL-6 > 4.53 pg/mL and TNF > 7.81 pg/mL as unfavorable risk factors.

**Table 1 life-11-01006-t001:** Baseline characteristics for the study population, stratified by outcome.

Variable	All Patients (*n* = 57)	Event-Free Group (*n* = 30)	Event Group (*n* = 27)	*p*
Clinical characteristics				
Age, years	47 (44–53)	48 (47–53)	46 (39–54)	0.19
Male gender, *n* (%)	45 (79%)	23 (79)	22 (79)	0.95
NYHA III-IV class, *n* (%)	51 (90%)	24 (83)	27 (96)	0.19
Body mass index, kg/m^2^	26.84 (23.4–31.6)	27.8 (22.7–32.4)	26.6 (23.6–30.4)	0.8
Systolic blood pressure, mmHg	114 (100–130)	123 (110–130)	106 (94–116)	<0.01
Diastolic blood pressure, mmHg	80 (70–80)	80 (70–80)	70 (69–80)	0.11
Atrial fibrillation, *n* (%)	11 (19)	6 (20)	5 (19)	0.89
Dyslipidemia, *n* (%)	10 (18)	6 (20)	4 (15)	0.73
Rheumatologic disease, *n* (%)	4 (7)	1 (3)	3 (11)	0.34
Duration of heart failure symptoms, months	12 (5–60)	10 (4–48)	48 (10–72)	0.08
Laboratory findings				
Hemoglobin, g/L	144 (130–151)	146 (137–154)	143 (126–150)	0.41
eGFR, mL/min/1.73 m^2^	85 (73–105)	83 (73–100)	93 (73–107)	0.27
BNP, ng/L	728 (90–1887)	248 (46–893)	1280 (343–2681)	<0.01
hs-troponin T, pg/mL	29.9 (18.5–48.5)	25.8 (16.2–45.1)	31 (22.7–59.7)	0.34
Concomitant cardiac medication				
ACE-I/ARB, *n* (%)	41 (72%)	21 (73)	19 (68)	0.71
Beta-blocker, *n* (%)	54 (95%)	28 (97)	26 (93)	0.61
MRA, *n* (%)	51 (90%)	24 (83)	27 (96)	0.19
Loop diuretics, *n* (%)	53 (93%)	26 (90)	27 (96)	0.61
Echocardiographic parameters				
LVEF, %	24 (2–32)	30 (21–35)	22 (18.8–26)	0.03
LV end-diastolic diameter, cm	6.9 (6.2–7.3)	6.7 (6.2–7.1)	6.9 (6.5–7.5)	0.17
Severe LV diastolic dysfunction, *n* (%)	23 (40)	7 (24)	16 (57)	0.01
Severe RV systolic dysfunction, *n* (%)	16 (28)	5 (17)	11 (39)	0.06
RV end-diastolic diameter, cm	3.3 (2.9–3.6)	3.1 (2.6–3.4)	3.6 (3.1–3.9)	<0.01
Hemodynamic measurements (*n* = 54)				
PCWP, mmHg	20 (15–30)	18 (14.8–24.8)	23 (16–34)	0.11
mPAP, mmHg	29 (21–39)	25 (21–37.3)	34 (27–43)	0.06
Cardiac index, L/min/m^2^	2.2 (1.6–2.8)	2.27 (1.93–2.81)	2.0 (1.5–2.49)	0.14

Values are expressed as: median (25th percentile, 75th percentile) or *n* (%). ACE-I—angiotensin-converting enzyme inhibitor, ARB—angiotensin II receptor blocker, BNP—B type natriuretic peptide, eGFR—estimated glomerular filtration rate, LV—left ventricle, LVEF—left ventricle ejection fraction, mPAP—mean pulmonary arterial pressure, MRA—mineralocorticoid receptor antagonist, NYHA—New York Heart Association, PCWP—pulmonary capillary wedge pressure, RV—right ventricle.

**Table 2 life-11-01006-t002:** Inflammation-related biomarkers of the study population, stratified by outcome.

Variable	All Patients (*n* = 57)	Event-Free Group (*n* = 30)	Event Group (*n* = 27)	*p*
Inflammation-related serum biomarkers				
WBC × 10^9^/L (*n* = 57)	7.92 (5.92–10.06)	8.33 (6.19–10.75)	7.83 (5.7–9.13)	0.48
hs-CRP, mg/L (*n* = 56)	4.6 (1.5–15.7)	2.4 (1.3–15)	6.6 (2.7–16.1)	0.27
IL-6, pg/mL (*n* = 55)	2.5 (4.7)	2.0 (2.0–3.8)	5.24 (2.0–13.7)	0.002
TNF-α, pg/mL (*n* = 55)	8.6 (6.6–10)	7.5 (6.1–9.3)	9.0 (7.9–12.9)	0.03
Adiponectin, μg/mL (*n* = 55)	14.2 (7.97–28.85)	10.9 (5.63–18.96)	23.4 (11.1–32.9)	0.01
suPAR, ng/mg protein (*n* = 45)	1.79 (0.84–2.65)	1.77 (0.92–2.84)	1.79 (0.79–2.65)	0.92
Inflammatory cells in myocardium (*n* = 55)				
CD3+, cells/mm^2^	10 (7–16)	10 (7–13)	10 (7–50)	0.39
CD45ro+, cells/mm^2^	7 (5–10)	7 (5–9)	6.5 (5–10.5)	0.71
CD68+, cells/mm^2^	4 (3–5)	4 (3–5)	3.5 (3–5)	0.96
CD4+, cells/mm^2^	4 (2–6)	4 (2–6)	3 (2–8)	0.71
CD54+, cells/mm^2^	0 (0–1)	0 (0–2)	0 (0–1)	0.29
HLA-DR+, cells/mm^2^	5 (4–6)	5 (4–6)	5 (4–7)	0.71

Values are expressed as: median (25th percentile, 75th percentile). hs-CRP—high-sensitivity C-reactive protein, IL-6—interleukin 6, suPAR—soluble urokinase-type plasminogen activator receptor, TNF-α—tumor necrosis factor α, WBC—white blood cell.

**Table 3 life-11-01006-t003:** Correlations between inflammatory-related biomarkers and parameters of heart failure severity.

	LVEF		mPAP		PCWP		BNP	
	r	*p*	r	*p*	r	*p*	r	*p*
WBC × 10^9^/L	−0.09	0.5	0.23	0.12	0.05	0.7	0.01	0.97
hs-CRP, mg/L	−0.3	0.03	0.19	0.19	0.17	0.2	0.33	0.02
IL-6, pg/mL	−0.56	<0.0001	0.48	<0.001	0.42	<0.01	0.66	<0.0001
TNF-α, pg/mL	−0.29	0.03	0.39	<0.01	0.25	0.08	0.5	<0.01
Adiponectin, μg/mL	−0.37	<0.01	0.43	<0.01	0.39	<0.01	0.65	<0.0001
suPAR, ng/mg protein	−0.07	0.66	0.2	0.21	0.11	0.48	0.25	0.11
CD3+, cells/mm^2^	−0.01	0.96	0.12	0.41	0.12	0.4	0.31	0.02
CD45ro+, cells/mm^2^	−0.04	0.78	0.06	0.67	0.03	0.84	0.28	0.04
CD68+, cells/mm^2^	−0.16	0.23	0.09	0.51	0.06	0.69	0.22	0.11
CD4+, cells/mm^2^	0.18	0.18	−0.03	0.81	0.02	0.9	−0.01	0.98
CD54+, cells/mm^2^	−0.1	0.7	0.1	0.57	0.12	0.39	0.15	0.28
HLA-DR+, cells/mm^2^	−0.19	0.16	−0.02	0.88	0.04	0.76	0.12	0.38

BNP—B type natriuretic peptide, hs-CRP—high sensitivity C-reactive protein, IL-6—interleukin 6, LVEF—left ventricle ejection fraction, mPAP—mean pulmonary arterial pressure, NYHA—New York Heart Association, PCWP—pulmonary capillary wedge pressure, suPAR—soluble urokinase-type plasminogen activator receptor, TNF-α—tumor necrosis factor α, WBC—white blood cell. NYHA—New York Heart Association.

**Table 4 life-11-01006-t004:** Correlations between inflammatory cells in the myocardium and inflammatory-related biomarkers in serum.

	CD3+, cells/mm^2^		CD45ro+, cells/mm^2^		CD68+, cells/mm^2^		CD4+, cells/mm^2^		CD54+, cells/mm^2^		HLA_DR+, cells/mm^2^	
	r	*p*	r	*p*	r	*p*	r	*p*	r	*p*	r	*p*
WBC × 10^9^/L	−0.12	0.42	0.1	0.52	0.12	0.23	−0.06	0.67	0.1	0.48	−0.14	0.35
hs-CRP, mg/L	0.19	0.18	0.23	0.12	0.26	0.07	0.003	0.99	0.01	0.97	0.12	0.39
IL-6, pg/mL	0.21	0.12	0.09	0.51	0.35	0.009	−0.22	0.11	−0.03	0.89	0.09	0.53
TNF-α, pg/mL	0.21	0.14	0.01	0.94	0.2	0.15	−0.07	0.64	−0.05	0.72	0.1	0.49
Adiponectin, μg/mL	0.14	0.3	0.11	0.42	0.05	0.72	−0.2	0.87	0.02	0.91	0.13	0.37
suPAR, ng/mg protein	0.08	0.62	0.16	0.3	0.13	0.42	0.19	0.23	0.1	0.52	0.09	0.55

hs-CRP—high sensitivity C-reactive protein, IL-6—interleukin 6, suPAR—soluble urokinase-type plasminogen activator receptor, TNF-α—tumor necrosis factor α, WBC—white blood cell.

**Table 5 life-11-01006-t005:** Results of Cox regression analysis for predictors of adverse cardiac events.

	Univariate		Multivariate	
Variable	HR (95% CI)	*p*	HR (95% CI)	*p*
Systolic blood pressure, mmHg	0.97 (0.95–0.995)	0.02		
Diastolic blood pressure, mmHg	0.96 (0.93–1.0)	0.05	0.95 (0.91–0.99)	0.01
LVEF, %	0.93 (0.88–0.97)	0.003		
Severe RV systolic dysfunction	2.81 (1.29–6.11)	0.009		
Severe LV diastolic dysfunction	3.15 (1.45–6.84)	0.003		
RV end-diastolic diameter, cm	2.65 (1.4–5.0)	0.003		
PCWP, mmHg	1.06 (1.01–1.11)	0.01		
mPAP, mmHg	1.05 (1.01–1.08)	0.01		
log BNP, ng/L	1.47 (1.16–1.86)	0.002		
hs-troponin T, pg/mL	1.004 (1–1.01)	0.068	1.005 (1.0–1.01)	0.04
IL-6 > 4.53 pg/mL	4.78 (2.09–10.89)	0.0002	6.26 (2.52–15.58)	<0.0001
hs-CRP > 4.62 µg/mL	2.57 (1.14–5.79)	0.02		
TNF-α > 7.81 pg/mL	3.42 (1.28–9.15)	0.01		
Adiponectin > 17.14 μg/mL	3.2 (1.43–7.16)	0.005	5.43 (1.01–10.33)	<0.001
CD3+ > 13 cells/mm^2^	2.18 (1.01–4.7)	0.048		
CD45ro+ > 11.5 cells/mm^2^	2.89 (1.22–6.87)	0.012	3.23 (1.01–10.33)	0.048

BNP—B type natriuretic peptide, CI—confidence interval, HR—hazard ratio, hs—high-sensitivity, hs-CRP—high-sensitivity C reactive protein, IL-6—interleukin 6, LV–left ventricle, LVEF—left ventricle ejection fraction, mPAP—mean pulmonary arterial pressure, PCWP—pulmonary capillary wedge pressure, RV—right ventricle, TNF-α—tumor necrosis factor α.

## Data Availability

The datasets during and/or analyzed during the current study available from the corresponding author on reasonable request.

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
