# Peer review of "Inflammation-Related Biomarkers Are Associated with Heart Failure Severity and Poor Clinical Outcomes in Patients with Non-Ischemic Dilated Cardiomyopathy"

_life, 2021, doi:10.3390/life11101006_

Round 1

Reviewer 1 Report

I read with interest the study by Kažukauskienė et al. regarding the predictive role of inflammatory biomarkers in a population of patients with non-ischemic dilated cardiomyopathy. 

The authors show that certain inflammatory biomarkers correlate with other heart failure severity parameters and predict clinical outcomes. Systemic inflammatory markers were more predictive of clinical outcomes than myocardial ones. The combination of 3 systemic biomarkers could identify low- and high-risk patients.

In an era of directed anti-inflammatory treatment in cardiovascular disease, this study provides an important observation in a specific group of NI-DCMP patients. 

I do have some comments:

1. Timing of biopsies- this should be stated in the Methods section. When was the biopsy obtained in relation to HF symptoms onset / diagnosis of NI-DCMP? Is there a difference between the groups from that standpoint? 

2. I feel the authors should emphasize and discuss the fact that the systemic inflammation biomarkers had a more impressive predictive value than the myocardial ones. This may imply that the outcomes are not necessarily driven by the cardiac inflammation itself. 

3. Cox regression- I am concerned that with only 27 events, using 15 variables may well result in overfitting of the model and limit its generalizability. 

4. As IL-1beta is considered a main driver of the inflammatory process and broadly studied in the context of cardiac involvement, I wonder why it was decided not to measure it as well.  

5. Table 1- Were there other co-morbidities? If so, please include them in the table. How was eGFR calculated? How was severe LV diastolic dysfunction defined and why was it not included in the regression model? 

6. Line 58- Ref. 9 claims the opposite. 

7. Methods- The fact that this is a retrospective analysis should be stated more clearly in the opening paragraph. 

8. There are some phrasings that need to be corrected:

Line 42- "or until..."

Lines 108-110- "Therefore..."

Line 329- and mPAP

Lines 329-331- "Besides... biomarkers".

Reviewer 2 Report

Dear Authors

It is a well-written and interesting complex study which analyses blood biomarkers, especially inflammatory biomarkers, immunohistochemistry of myocardial biopsy specimens, and hemodynamics data of patients with heart failure due to nonischemic cardiomyopathy.

Patients were stratified in two groups  by the presence of a composite major end point during five years follow-up (death, heart transplant, or LVAD implantation).

That is a continuation of the work of authors on the same group of patients which were described  in Cardiol J 408 2020;XX, No. X:Epub ahead of print. https://doi.org/10.5603/cj.a2020.0088 (reference 13) where finding of the presence of viral genome  and myocarditis in myocardial biopsies with a similar composite end point was a main goal. Interestingly but maybe a little unexpectedly with a neutral outcome on survival between different groups of patients – maybe this is the effect of low number of pts.

A separate substudy publication in a subgroup of pts (41) concentrated on ultrasound characteristics  in relation to hemodynamics of the pulmonary circulation was described by the authors in Cardiovasc Ultrasound 2021;19:21. https://doi.org/10.1186/S12947-021-00254-1. (ref 15).

The study is important for the understanding of the possible role of inflammatory markers in risk assessment, however in my opinion it needs some additional work

My main critical remarks are:

  1. The group Is very well described with very few missing data – however rather small- it makes the cut-off points of markers rather not easily generalized for other non-ischemic cardiomyopathy pts.
  2. Did you find a correlation between myocardial biopsy inflammation findings and blood sample inflammation markers (please add a Table like table 3)
  3. Probably due to the relatively small number of pts (57) Authors were forced to use a composite end point which has its components not equivalent :death and OHT (especially when planned OHT is included are a different spectrum of events)
  4. Do you include only urgent heart transplantation or also planned heart transplantation. LVAD -what INTERMACS class please specify

4 . In similar situations, a concurring risk model is used (please see ISHLT report publications).

  1. What was the distribution of separated end points (Death, LVAD , OHT between both groups?
  2. Did you try to analyze separately KM curves treating non -urgent OHT as censoring events?
  3. The construction of the study determines that it can be used only as generating hypothesis study -please be cautious about its practical implications.

Minor remarks

Page 9 line 243 should probably be hs-CRP

Round 2

Reviewer 2 Report

Dear Authors

I thank you for your clarification and revision of the manuscript.

My main concerns were addressed and clarified to some extent.

The drawbacks of composite end points are well known and I agree that they are used in many studies. My concern was that urgent transplantation (1 or 2 UNOS status) or urgent LVAD (low INTERMACS status) can be treated to some extent as a similar outcome as death. However, this is not the case for planned heart transplant or LVAD at INTERMACS 4 or higher. They should not be treated equally to death. Some patients on the active transplant list despite poor clinical status can live even 2 or 3 years without transplant. It is especially important for markers that are supposed to pinpoint the true risk of the patient.  Fortunately, the number of planned transplants  or high LVAD status lNTERMACS is rather small in your data.

I suggest that the results of analyses from point 6 and KM curves are included in the additional material.

  1. see your response to my remark: Did you try to analyze separately KM curves treating non -urgent OHT as censoring events?

As this is a small sample study, every patient is extremely important for analysis. Censoring non-urgent events would mean "losing" these patients, reducing the power of results, as well as reducing patients' number in the event group and increasing patients number in event-free group.

However, out of curiosity, we performed Kaplan-Meier analysis with censored 4 nonurgent patients. The difference between survival curves remained similar when patients were stratified by the cut-offs of hs-CRP or IL-6. However, when the cut-offs of adiponectin, CD3+ and CD45ro+ stratified the patients, the difference between the curves increased, while 3 or 4 censored patients tended to have a lower number of cells and adiponectin concentration. Contrarily, all censored patients tended to have higher concentrations of TNF- α; therefore the difference between the curves decreased.

Clarifying point 5 of my review: In a similar situation with different outcomes, a more complicated model is often used. See a clarifying publication Eur Heart J. 2014 Nov 7; 35(42): 2936–2941.

Published online 2014 April 7. doi: 10.1093/eurheartj/ehu131

However, in future analyses, I strongly suggest to not concatenate planned heart transplantation and LVAD implantation with high INTERMACS status in the same end point.
